# Metasurface-assisted phase-matching-free second harmonic generation in lithium niobate waveguides

Cheng Wang[1], Zhaoyi Li[2], Myoung-Hwan Kim[3], Xiao Xiong[1,4], Xi-Feng Ren[4], Guang-Can Guo[4], Nanfang Yu[2] & Marko Lončar[1]

The phase-matching condition is a key aspect in nonlinear wavelength conversion processes, which requires the momenta of the photons involved in the processes to be conserved. Conventionally, nonlinear phase matching is achieved using either birefringent or periodically poled nonlinear crystals, which requires careful dispersion engineering and is usually narrowband. In recent years, metasurfaces consisting of densely packed arrays of optical antennas have been demonstrated to provide an effective optical momentum to bend light in arbitrary ways. Here, we demonstrate that gradient metasurface structures consisting of phased array antennas are able to circumvent the phase-matching requirement in on-chip nonlinear wavelength conversion. We experimentally demonstrate phase-matching-free second harmonic generation over many coherent lengths in thin film lithium niobate waveguides patterned with the gradient metasurfaces. Efficient second harmonic generation in the metasurface-based devices is observed over a wide range of pump wavelengths ($\lambda = 1580$–$1650$ nm).

[1] John A. Paulson School of Engineering and Applied Sciences, Harvard University, Cambridge, MA 02138, USA. [2] Department of Applied Physics and Applied Mathematics, Columbia University, New York, NY 10027, USA. [3] Department of Physics, The University of Texas Rio Grande Valley, Brownsville, TX 78520, USA. [4] Key Laboratory of Quantum Information & Synergetic Innovation Center of Quantum Information & Quantum Physics, University of Science and Technology of China, Hefei, Anhui 230026, China. Cheng Wang and Zhaoyi Li contributed equally to this work. Correspondence and requests for materials should be addressed to N.Y. (email: ny2214@columbia.edu) or to M.Lča. (email: loncar@seas.harvard.edu)

Nonlinear optics has a wide range of important applications, including frequency conversion[1-3], quantum light sources[4,5], optical frequency combs[6], and ultrafast all-optical switches and memories[7,8]. To achieve efficient nonlinear optical processes, the phase-matching condition has to be strictly satisfied[9]. This ensures that generated nonlinear optical signals are added constructively, and optical power is transferred continuously from the pump(s) to the signal. In the case of bulk nonlinear media, phase matching can be achieved by using crystal birefringence, by controlling the angle of intersection between optical beams, or by using periodically poled nonlinear crystals (i.e., quasi-phase matching)[9].

Nonlinear effects can be significantly enhanced inside nanophotonic waveguides with tight light confinement[10-17]. Nanophotonics also provides alternative tools to achieve phase-matching; examples include dispersion-engineered structures[10-13], photonic crystal waveguides[14,15], anisotropic micro-cavities[16], and periodically poled nonlinear waveguides[17]. Nevertheless, the nonlinear phase-matching condition still needs to be satisfied between co-propagating light waves.

Strict phase-matching is not required in resonant structures with (sub-)wavelength-scale mode volumes, where the conversion efficiency relies on modal overlap between fundamental modes and higher harmonics[18-32]. For example, doubly resonant photonic cavities with high quality factors have been proposed to achieve highly-efficient wavelength conversion, while their narrow bandwidth remains a major challenge in experiments[18-21]. Plasmonic nanoantennas have also been used to achieve enhanced optical nonlinearities[22-27]. Simultaneous field enhancement at both the fundamental and harmonic wavelengths and spatial overlap between the modes allow for even higher conversion efficiencies[28,29]. More recently, dielectric antennas have emerged as a promising alternative over plasmonic ones due to their lower optical loss and higher damage threshold. Silicon Mie resonators supporting magnetic dipole resonances[30] and Fano resonances[31] have been utilized to enhance third harmonic generation. However, these devices suffer from low overall conversion efficiencies because light–matter interactions occur within a limited volume of nonlinear media.

Here, we theoretically propose and experimentally demonstrate a hybrid nonlinear integrated photonic device consisting of phase-gradient metasurfaces patterned on top of a nonlinear waveguide. Our devices leverage the high nonlinear susceptibility of lithium niobate (LiNbO$_3$, or LN)[33], the strong capability of gradient metasurfaces in controlling the modal indices of waveguide modes[34], and the large volume of light–matter interaction provided by optical waveguides. We show that the unidirectional effective wavevector provided by gradient metasurfaces enables a one-way transfer of optical power from the pump to the second harmonic (SH) signal, which represents a unique scheme of phase-matching-free nonlinear generation, where the nonlinear generation efficiency is not sensitive to the variation of pump frequency and device geometry. We demonstrate efficient, broadband, and robust second harmonic generation (SHG), where the SH signal monotonically increases over many coherence lengths inside LN waveguides patterned with gradient metasurfaces.

## Results

**Theoretical principle of the phase-matching-free process.** Figure 1a-c show the schematic views and scanning electron microscope (SEM) images of our devices, where gradient metasurface (a phased array of dielectric antennas) is patterned on the

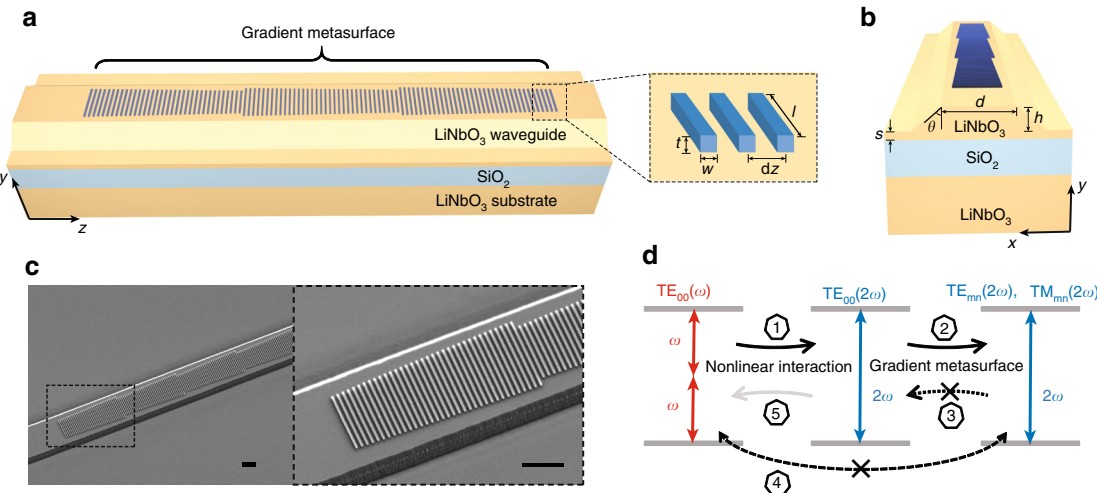

**Fig. 1** Device schematics and working principle. **a** Schematic of an integrated nonlinear photonic device, where a gradient metasurface consisting of arrays of dielectric phased antennas is used to achieve phase-matching-free second harmonic generation (SHG) in a LiNbO$_3$ waveguide. For clarity, only three antenna arrays are shown, but there could be more than three arrays. Inset shows a magnified view of the antenna array. **b** Schematic of the device cross-section. **c** Left: Scanning electron microscope (SEM) image of a fabricated device showing four phased antenna arrays consisting of silicon nano-rods of different lengths patterned on the top surface of a LiNbO$_3$ waveguide. Right: Zoom-in view of the device (dashed frame in the left image). Scale bar: 1 μm for both images. **d** Conceptual diagram of the proposed phase-matching-free SHG. The optical power is first coupled from the pump, TE$_{00}$($\omega$), to the fundamental waveguide mode at the second harmonic (SH) frequency, TE$_{00}$($2\omega$) (arrow #1 in the diagram), and then with the assistance of the gradient metasurface, coupled to the higher-order waveguide modes at the SH frequency, TE$_{mn}$($2\omega$) and TM$_{mn}$($2\omega$) (arrow #2). The unidirectional wavevector provided by the gradient metasurface ensures that coupling of optical power back from the TE$_{mn}$($2\omega$) and TM$_{mn}$($2\omega$) modes to the TE$_{00}$($2\omega$) mode is highly inefficient (arrow #3). The coupling of optical power from the TE$_{mn}$($2\omega$) and TM$_{mn}$($2\omega$) modes to the pump, TE$_{00}$($\omega$), is also highly inefficient (arrow #4), because the spatial overlap between TE$_{mn}$($2\omega$) or TM$_{mn}$($2\omega$) and TE$_{00}$($\omega$) in the waveguide cross-section is small (i.e., coupling coefficient is small). In this way, asymmetric optical power transfer is realized between the pump and the SH signal (i.e., process indicated by arrow #1 is more efficient than the process indicated by arrow #5) and as a result the SHG power monotonically increases as a function of the propagation distance. This new scheme circumvents the nonlinear phase-matching condition as long as the gradient metasurface provides an asymmetric effective wavevector to preferentially couple optical power from lower to higher-order waveguide modes at the SH frequency and does not interact with the pump

top surface of a nonlinear optical waveguide. We utilize the dipolar Mie resonances in dielectric nano-rod antennas with different lengths to introduce different phase shifts in the scattered light waves[34]. Collectively, the phased antenna array creates a unidirectional phase gradient $d\Phi/dz$, which is equivalent to a unidirectional effective wavevector $\Delta k$, along the waveguide. Here, $d\Phi$ is the difference in phase response between adjacent nanoantennas that are separated by a subwavelength distance of $dz$. The unidirectional effective wavevector $\Delta k$ enables directional coupling of waveguide modes. That is, when an incident waveguide mode propagates against the direction of $\Delta k$, the modal index decreases, which corresponds to coupling into higher-order waveguide modes; the inverse process in which optical power is coupled from the higher-order modes to the lower-order ones is prohibited due to the lack of a phase-matching mechanism.

Figure 1d shows the working principle of the metasurface-based nonlinear integrated photonic devices. In the region of the nonlinear waveguide patterned with the metasurface structure, once optical power couples from the fundamental mode at the pump frequency, $TE_{00}(\omega)$, to the fundamental mode at the SH frequency, $TE_{00}(2\omega)$, it immediately starts to be converted into higher-order waveguide modes at the SH frequency, $TE_{mn}(2\omega)$ and $TM_{mn}(2\omega)$, by the gradient metasurface (Supplementary Fig. 1). The unidirectional wavevector provided by the gradient metasurface ensures that optical power cannot be coupled from higher-order modes, $TE_{mn}(2\omega)$ and $TM_{mn}(2\omega)$, back to $TE_{00}(2\omega)$ (dotted arrow in Fig. 1d). Furthermore, the optical power carried by $TE_{mn}(2\omega)$ and $TM_{mn}(2\omega)$ cannot be coupled back to the pump, $TE_{00}(\omega)$ (dashed arrow in Fig. 1d), because the coupling coefficient between them is negligible (i.e., spatial overlap between $TE_{mn}(2\omega)/TM_{mn}(2\omega)$ and $TE_{00}(\omega)$ on the waveguide cross-section is very small). In this way, optical power is retained in the SH signal and accumulates as a function of propagation distance. The antennas are designed to interact with guided waves at the SH frequency, and they are too small to strongly scatter the pump. As a result, the pump propagates as the fundamental waveguide mode through the device with decreasing power, while the SH signal increases monotonically in the nonlinear waveguide patterned with the gradient metasurface. The order of waveguide modes at the SH frequency, however, keeps increasing; as such, the SH signal will eventually leak out from the waveguide when the cutoff condition for waveguiding is reached. This sets the ultimate limitation on the highest nonlinear conversion efficiency achievable in the current device scheme.

**Device design and numerical simulation**. We choose lithium niobate (LN) as the nonlinear waveguide material. For decades, LN has been the most widely used material for nonlinear generation processes owing to its broad transparency window ($\lambda = 400$ nm–5 μm) and large second-order nonlinear susceptibility ($d_{33} = 27$ pm/V)[33]. Conventionally, LN waveguides are formed by using metal in-diffusion or ion exchange to slightly increase the refractive index of the waveguide core ($\Delta n \sim 0.02$)[1,35]. Recently, the platform of LN on insulator (LNOI) has emerged as a promising candidate for next-generation on-chip wavelength conversion. This platform is based on sub-micron thick LN thin films bonded to an insulator (silica) substrate using a smart-cut technique, resulting in devices with much increased index contrast ($\Delta n > 0.6$) and reduced modal size[36]. Optical devices with excellent wavelength-scale optical confinement and efficient nonlinear generation capability have been realized in the LNOI platform[12,17,37–39].

We choose amorphous silicon (a-Si) as the material for the dielectric phased array antennas. A-Si has a refractive index ($\sim 4$ in the visible) significantly higher than LN ($\sim 2.2$ in the visible) (Supplementary Fig. 2). Therefore, there is a strong interaction

between waveguide modes at the SH frequency and Mie resonance modes in the a-Si nanoantennas. The use of dielectric antennas instead of plasmonic antennas minimizes the absorption losses (Supplementary Fig. 3).

In our devices, the LNOI has a single crystal x-cut LN device layer with a thickness of 400 nm. A 2-μm buried $SiO_2$ layer is underneath the LN device layer. The ridge LN waveguides created by a reactive ion etching process has a trapezoidal cross-section with a width $d = 2600$ nm on the top, a sidewall tilting angle $\theta = 40$ degree, a ridge height $h = 300$ nm and an underetched slab thickness $s = 100$ nm (Fig. 1b). A gradient metasurface consisting of a number of identical phased antenna arrays is patterned along the center of the top surface of the LN ridge waveguide. Each antenna array consists of 35 a-Si nanoantennas with a thickness $t = 75$ nm, a width $w = 75$ nm and a range of lengths $l$ (Fig. 1a, inset). The separation, $dz$, between adjacent nanoantennas is 140 nm and the phase difference, $d\Phi$, between them is 0.5 degree at the SH frequency (i.e., $\lambda \sim 750$ nm). Note that the actual value of $d\Phi$ is not critical for the phase-matching-free process to take effect. As long as the antenna array converts SH light into higher-order waveguide mode(s) at a rate comparable to the SHG process, the total SH signal will keep accumulating.

Figure 2a shows simulated performance of our devices in comparison to that of a bare nonlinear waveguide. Detailed numerical simulation methods are provided in Supplementary Fig. 2. In a bare LN waveguide, the generated SH signal is mostly carried by the $TE_{00}(2\omega)$ mode because of the efficient nonlinear overlap between the two fundamental waveguide modes, $TE_{00}(2\omega)$ and $TE_{00}(\omega)$ (i.e., integration of $TE_{00}^*(2\omega) \times TE_{00}^2(\omega)$ over the waveguide cross-section is significantly larger than other mode combinations). However, due to the large phase mismatch between the two modes, optical power is frequently exchanged between them along the waveguide (with a period of twice the coherent buildup length $\sim 2$ μm, in our case; oscillatory black curves in Fig. 2a); thus, the SH signal can never reach a high intensity. However, when the nonlinear waveguide is patterned with the gradient metasurface structure, the SHG power monotonically increases as a function of the propagation distance, and a longer metasurface structure consisting of more sets of phased antenna arrays produces SH signals with higher intensities. For example, using just one set of phased antenna array (4.76 μm in length), the SHG power is increased by seven times compared with the peak value achievable in a bare waveguide. Using six sets of phased antenna arrays with a total length of 28.6 μm, the SHG power can reach a value two orders of magnitude higher than that achievable in a bare waveguide (Fig. 2a). Increasing the number of phased antenna arrays can further improve the nonlinear conversion efficiency, until a point at which the generated SH signal is coupled by the gradient metasurfaces into leaky waves. Our simulations show that with the current device configuration, a monotonic increase of SHG power could be sustained for at least 11 sets of phased antenna arrays (i.e., 54 μm in length).

Figure 2b shows the mode evolution at both the pump and SH frequencies as a function of the propagation distance in a waveguide section patterned with three sets of phased antenna arrays. The simulation results show that the pump power is primarily carried by the fundamental waveguide mode, $TE_{00}(\omega)$, throughout the metasurface-patterned region, indicating a weak interaction between the pump and the metasurface structure. At the SH frequency, the x-component of the electric field has mainly a single lobe, which indicates that optical power is converted from the $TE_{00}(\omega)$ mode to the $TE_{00}(2\omega)$ mode. The electric field polarized along the y-axis at the SH frequency shows multiple lobes, and this is the result of coupling of optical power from the $TE_{00}(2\omega)$ mode to higher-order $TE_{mn}(2\omega)$ ($TM_{mn}(2\omega)$) modes by the gradient metasurface. The interference between

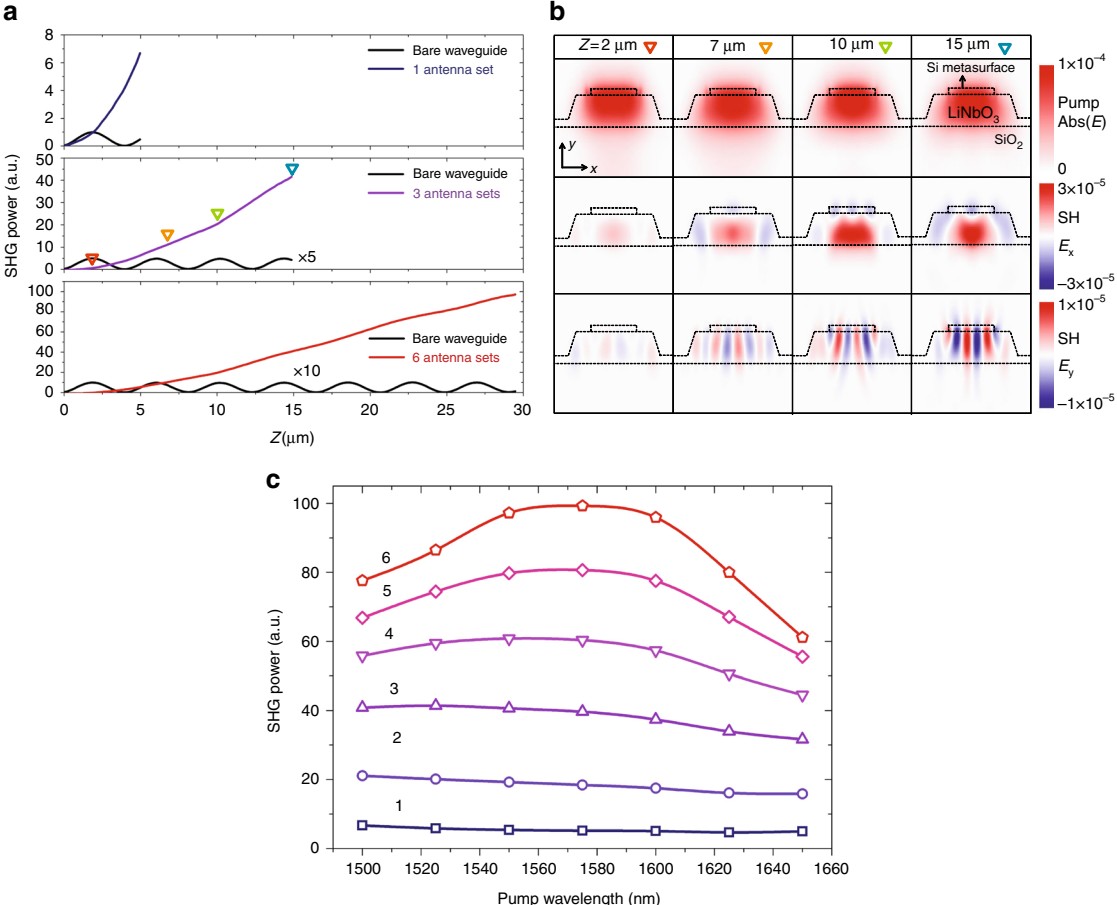

**Fig. 2** Numerical simulation. **a** Simulations showing that the second harmonic (SH) signal increases monotonically as the number of phased antenna arrays increases. Without the antenna arrays, the SH signal oscillates periodically along the waveguide (black curves). The colored triangles in the middle panel indicate the positions of the simulated cross-section profiles shown in **b**. **b** Simulations showing the evolution of the pump and the SH signal as they propagate along the waveguide. It can be seen that the pump stays in the fundamental waveguide mode, $TE_{00}(\omega)$ (first row), while the SH is gradually coupled into higher-order waveguide modes by the gradient metasurface (second and third row). **c** Simulations showing that the SH power increases as the number of the phased antenna arrays (labeled 1–6) increases and that the second harmonic generation (SHG) process is broadband (i.e., efficient SHG is observed over a wide range of pump wavelengths). The broadband performance is achieved because the unidirectional effective wavevector provided by the gradient metasurface breaks the symmetry of coupling between the pump and the SH signal so that optical power is transferred preferentially from the pump to the SH signal while the inverse process is highly inefficient. The discrete symbols represent numerical simulation results, and are connected using spline function to better illustrate the results

these higher-order modes after the metasurface-patterned section will cause intensity variations of the generated SHG signal, but the overall SHG power would stay on the same level as that right after the antenna arrays (Supplementary Fig. 4).

The asymmetric coupling between the pump and the SH signal as a result of the unidirectional wavevector provided by the gradient metasurface makes the SHG process tolerant to the variation of the pump frequency and the geometry of the device. Figure 2c shows the simulated SHG power as a function of the pump wavelength ranging from 1500 to 1650 nm, for different sets of phased antenna arrays. In the case of one set of antenna array, the generated SHG power is almost a constant for different pump wavelengths. The SHG enhancement bandwidth decreases, however, as the number of antenna arrays increases, since a longer interaction distance corresponds to less tolerance on the phase gradient of the antenna arrays. Nonetheless, for a device with six antenna arrays, the wavelength range within which the SHG power is above 80% of the peak value is still larger than 115 nm. In addition, our numerical simulation shows that the SHG process is robust against the variation of the device geometry. The change to the SHG power in a device with three sets of phased

antenna arrays is small when the lengths of the nano-rod antennas deviate from their designed values by ±10%, when the antenna arrays are offset from the center of the waveguide up to 100 nm, and when the width of the LN waveguide deviates from its designed values by ±100 nm (Supplementary Fig. 5). In comparison, in conventional schemes of nonlinear phase matching, the nonlinear wavelength conversion process is typically sensitively dependent on the optical alignment, pump wavelength and operating temperatures[9].

**Experimental characterization**. The devices were fabricated using a combination of electron-beam lithography, reactive ion etching and plasma-enhanced chemical vapor deposition (Fig. 3). The fabricated devices were characterized using a butt-coupling setup, where the telecom pump light with tunable wavelengths, $\lambda_{pump}$, was coupled into the devices using a tapered lensed fiber, and the generated SH signal was coupled out from the polished facets of the LN waveguides and collected using another tapered lensed fiber. Details on device fabrication and characterization can be found in Methods section.

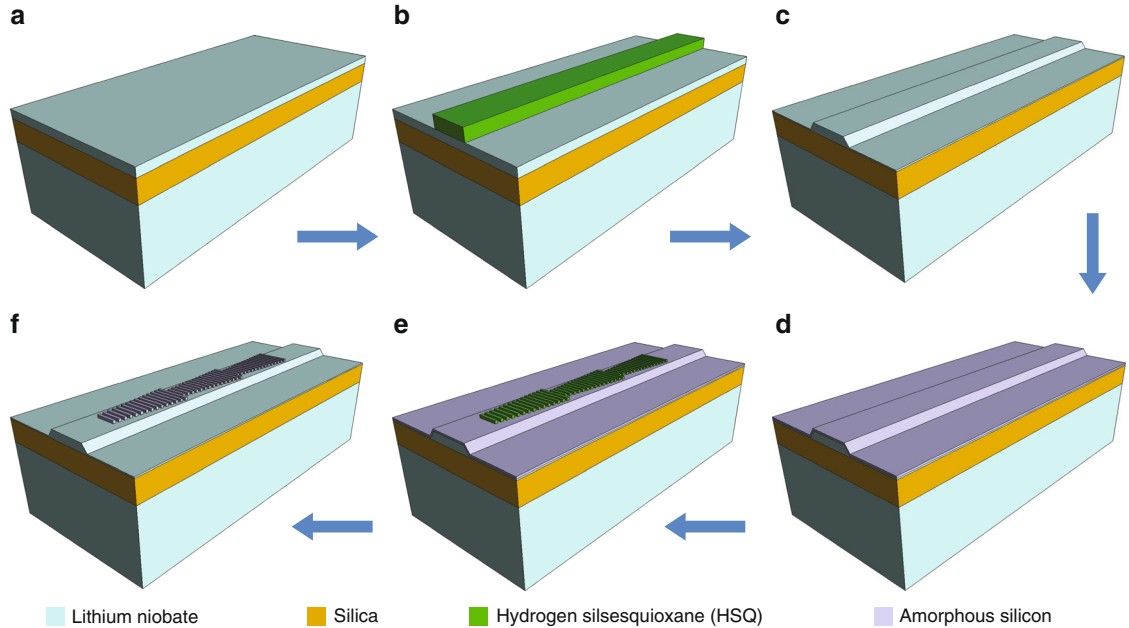

Lithium niobate    Silica    Hydrogen silsesquioxane (HSQ)    Amorphous silicon

**Fig. 3** Device fabrication steps. **a** The fabrication starts with an x-cut LiNbO3 thin film (400 nm thick) bonded on top of SiO$_2$. **b** Hydrogen silsesquioxane (HSQ) resist is spin-coated and patterned with electron-beam lithography (EBL). **c** LiNbO$_3$ ridge waveguides are formed using an optimized Ar$^+$ plasma etching technique. The residual resist is removed in buffered oxide etch (BOE). **d** A 75 nm thick p-doped amorphous silicon layer is deposited using plasma-enhanced chemical vapor deposition. **e** A second EBL process is performed to define the antenna patterns on top of the amorphous silicon surface. **f** Antenna arrays are created using reactive ion etching, before a second BOE etch is used to remove the residue HSQ

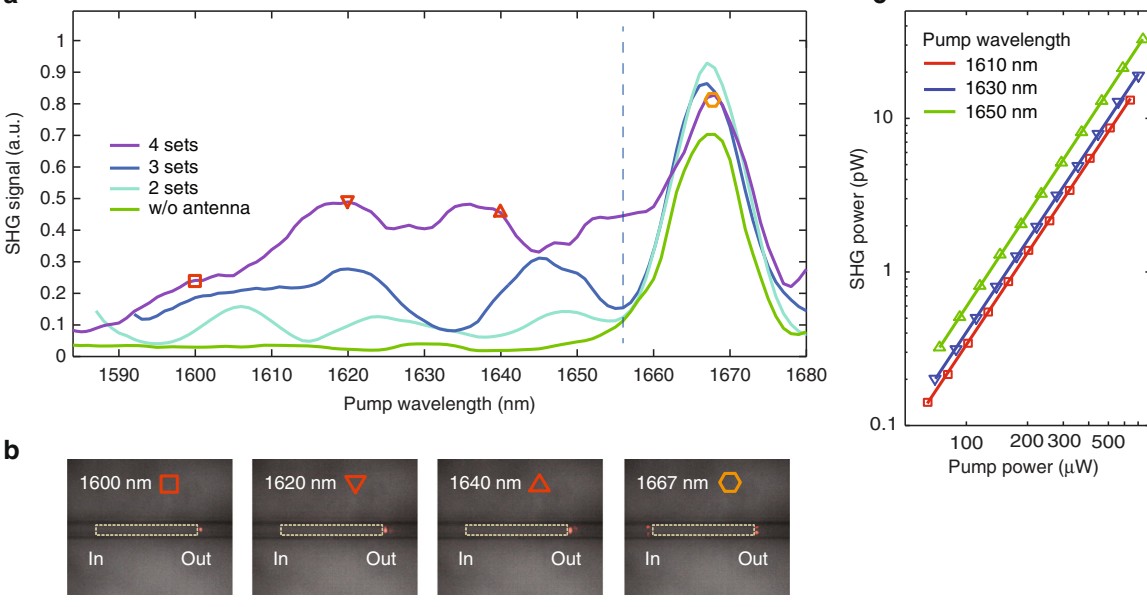

**Fig. 4** Experimental results. **a** Measured SHG power at the output port of devices patterned with different number of phased antenna arrays, in comparison with that of a bare LiNbO$_3$ waveguide, showing broadband SHG enhancement (to the left of the dashed line). The narrow SHG peak at a pump wavelength of 1667 nm (to the right of the dashed line), which exists in all tested devices including bare LiNbO$_3$ waveguides, is the result of an accidental phase-matching between the pump, TE$_{00}(\omega)$, and the seventh-order TE waveguide mode at the SH frequency, TE$_{06}(2\omega)$. **b** Top-view camera images showing the scattered SH light from a device patterned with a gradient metasurface consisting of four phased antenna arrays (indicated by the white dashed frames). At pump wavelengths of 1600, 1620, and 1640 nm, SH signal (red spots) is only observed at the output end of the gradient metasurface, which indicates that the SH signal is generated within the metasurface-covered section of the device. In contrast, at the accidental phase-matching peak wavelength of 1667 nm, scattered SH light appears at both the input and output ends, which indicates that the SH signal has already been generated by the phase-matching process before it enters the metasurface section. **c** Input–output power relations for the device showing quadratic power dependence between the pump and the SH signal at three different pump wavelengths

Figure 4a shows the measured SHG power from several devices that have the same LN waveguide patterned with different sets of phased antenna arrays, as well as SHG power produced by a bare control waveguide. Over the range of $\lambda_{pump} = 1580$–$1680$ nm, the SHG signal in all devices with the antenna arrays shows significant enhancement compared with the bare LN waveguide. The enhancement of the SHG process increases with increasing number of antenna arrays (Fig. 4a), which is in good agreement with the simulation results shown in Fig. 2. To further confirm that the SHG enhancement is indeed contributed by the antenna arrays, a visible camera is used to visualize the SH light scattered from the top of the devices. The SH signal is most likely to be scattered at the boundaries (labeled "input" and "output" in Fig. 4b) between the metasurface-patterned section of the LN waveguide and sections of the bare LN waveguide due to abrupt modal index changes. At three arbitrarily chosen pump wavelengths $\lambda_{pump} = 1600$, 1620, and 1640 nm, the scattered SH light is observed only at the output end of the antenna arrays, but not at the input end (Fig. 4b), indicating that the SH signal is generated within the metasurface-patterned section. Figure 4c shows the input–output power relations for a device patterned with four phased antenna arrays. Quadratic power dependence is observed at three pump wavelengths, indicating a broadband second-order nonlinear optical process. The measured conversion efficiencies for the three wavelengths are $3.2 \times 10^{-5}$, $3.8 \times 10^{-5}$, and $6.0 \times 10^{-5}$ W$^{-1}$. Considering the short metasurface-covered device length of 19 µm, the normalized conversion efficiencies are 890 % W$^{-1}$ cm$^{-2}$, 1050 % W$^{-1}$ cm$^{-2}$, and 1660 % W$^{-1}$ cm$^{-2}$, respectively, significantly higher than conventional PPLN and recent reports on thin film LN waveguides due to the strong nonlinear modal overlap in our devices[1,12,17].

Note that there is a SHG peak at $\lambda_{pump} = 1667$ nm for all tested devices, including bare LN waveguides (Fig. 4a). This is the result of an accidental phase-matching[12] between the $TE_{00}(\omega)$ mode and the $TE_{06}(2\omega)$ mode for our LN waveguide dimensions (Supplementary Fig. 6). At this peak wavelength, scattered SH light is observed at both the input and output ends of the antenna arrays (Fig. 4b), indicating that the SH signal has already been generated before light interacts with the antenna arrays. The measured SHG efficiency for this accidental phase-matching peak is ~$9 \times 10^{-5}$ W$^{-1}$, comparable to that of the broadband enhanced SH signal from the device with four phased antenna arrays, but is the result of coherent SHG accumulation over a waveguide total length of 1.5 mm, resulting in a normalized conversion efficiency of 0.4% W$^{-1}$ cm$^{-2}$. This indicates that, within the same nonlinear interaction length, the SHG process assisted by the gradient metasurfaces is at least three orders of magnitude more efficient than the accidental phase-matching process in a bare LN waveguide.

The measured SHG spectra are not as flat as those in simulations (Fig. 4a), which is likely due to two reasons. First, after light propagates through the metasurface-patterned region, there is a phase-mismatched interaction between the SH signal and the residual pump, which results in a small oscillation of the SH intensity as a function of the propagation distance (Supplementary Fig. 4). The spatial oscillation of the SH intensity and the dependence of the oscillation period on the pump wavelength lead to a variation of the SH signal measured at the output port of the LN waveguide as a function of the pump wavelength. Second, the SH signal is carried by a different combination of higher-order waveguide modes at different pump wavelengths, which leads to different overall collection efficiencies by the tapered lensed fiber.

## Discussion

In conclusion, we have demonstrated a new scheme of phase-matching-free nonlinear wavelength conversion that is based on the integration of gradient metasurface structures and nonlinear waveguides. The gradient metasurfaces break the symmetry of coupling between the pump and nonlinear signals so that the nonlinear generation process can maintain a high efficiency over a broad range of pump wavelengths. Furthermore, in contrast to antenna-based nonlinear nanophotonic devices reported in the literature where light–matter interactions occur within a limited volume of nonlinear materials[22–29], the collective effect of phased arrays of nanoantennas in the gradient metasurfaces enables us to utilize a significantly larger volume of nonlinear materials with dimensions many times of the pump wavelength to increase the nonlinear generation efficiency. We fabricated such devices by patterning phased arrays of a-Si nano-rod antennas on LN waveguides, and characterized their SHG properties. The measurement results show orders of magnitude enhancement in the nonlinear wavelength conversion process over a broad range of pump wavelengths. One drawback of our current approach is the difficulty of efficient light collection from the generated higher-order waveguide modes. This could possibly be solved by using a single antenna array to convert the SH light into one specific mode (e.g., $TM_{00}$)[34] and extracting the remaining pump light into an adjacent waveguide to repeat the process.

## Methods

**Numerical simulation**. Nonlinear optical simulations were performed using FDTD methods (Lumerical). In our design, the width (75 nm), height (75 nm) and the center-to-center distance (140 nm) of the amorphous silicon nanoantennas were chosen according to our fabrication capabilities and kept constant. The phase response of the nanoantennas is obtained from FDTD simulations: a nano-antenna is placed on a LN substrate, a plane wave is incident onto the antenna from the substrate, and the phase of the scattered light from the antenna is monitored. The scattering phase monotonically increases as a function of antenna length. The gradient metasurface is created by assembling an array of nanoantennas where the phase difference between adjacent elements is 0.5°. Details on the complex optical refractive indices and LN crystal orientation used in FDTD simulations, and calculated antenna phase response can be found in Supplementary Fig. 2.

**Device fabrication**. X-cut LNOI wafers obtained from NANOLN, with a 400-nm thick device layer bonded on top of a 2-µm thick silica buffer layer, were used for device fabrication (Fig. 3). Hydrogen silsesquioxane (HSQ) resist (~600 nm) was spun on the wafers and patterned with electron-beam lithography (EBL). The patterned HSQ was subsequently used as an etching mask to define LN ridge waveguides using an optimized Ar$^+$ plasma etching technique[12,37]. The LN ridge waveguides have a top width of 2.6 µm and a height of 300 nm, leaving a 100-nm thick LN slab underneath. The residue HSQ was removed in buffered oxide etch (BOE). A 75 nm thick p-doped amorphous silicon (a-Si) layer was then deposited on top of the entire sample surface using plasma-enhanced chemical vapor deposition (PECVD). A second HSQ resist layer (~300 nm) was then patterned on the a-Si surface using EBL. Reactive ion etching (RIE) was performed to transfer the second HSQ pattern into the a-Si layer, defining the antenna arrays. The antennas have a width of 75 nm and the gap between adjacent antennas is 65 nm. Finally, a second BOE etch was used to remove the residue HSQ resist.

**Optical measurement**. Devices were characterized using a butt-coupling setup. Telecom pump light from a continuous wave tunable laser (Santec TSL-510, max power ~ 20 mW) was coupled into the LN waveguides through a tapered lensed fiber. A fiber polarization controller was used to ensure TE mode input. SH light was collected from the output waveguide facet using a second tapered lensed fiber and measured using a silicon avalanche photodiode. The scattered SH light was monitored by placing a visible CCD camera above the device, focused on the metasurface section. The input and output power used for conversion efficiency calculation refer to the on-chip power into and out of the metasurface region, taking into consideration the fiber-to-chip coupling losses. The normalized conversion efficiency is defined as $\eta = P_{out} \cdot P_{in}^{-2} \cdot L^{-2}$, where $P_{in}$ and $P_{out}$ are the input and output power, respectively, and $L$ is the length of the metasurface region.

**Data availability**. The data that support the findings of this study are available within the article and the Supplementary Information file.

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

## Acknowledgements

The work was supported in part by National Science Foundation (grant No. ECCS-1609549 and No. ECCS-1307948), the Air Force Office of Scientific Research (grant No. FA9550-14-1-0389 through a Multidisciplinary University Research Initiative program), Defense Advanced Research Projects Agency Young Faculty Award (grant No. D15AP00111), National Natural Science Foundation of China (grant No. 61590932), and the Open Fund of the State Key Laboratory on Integrated Optoelectronics (grant No. IOSKL2015KF12). Device was fabricated at the Center for Nanoscale Systems (CNS) at Harvard University. Research was carried out in part at the Center for Functional Nanomaterials, Brookhaven National Laboratory, which is supported by the U.S. Department of Energy, Office of Basic Energy Sciences, under contract no. DE-SC0012704.

## Author contributions

C.W., Z.L., M.-H.K., N.Y., and M.L. proposed the device schematic and conceived the experiment. Z.L. and M.-H.K. performed numerical simulations. C.W. fabricated the devices. C.W. and X.X. carried out the experiments. X.-F.R. and G.-C.G provided valuable support and feedback to the project. C.W., Z.L., N.Y., and M.L. wrote the manuscript with contribution from all authors. N.Y. and M.L. supervised the project.

## Additional information

**Competing interests:** The authors declare no competing financial interests.

