## [Peer Review File · Nature Communications]

Reviewers' comments:

Reviewer #1 (Remarks to the Author):

In this manuscript, the authors presented both theoretical and experimental research on second harmonic generation in a lithium niobate waveguide assisted with phase-gradient metasurface. A new mechanism of efficient second-order nonlinear optical interaction is demonstrated relying on mode conversions in the phase-gradient dielectric antenna array. This is an exciting progress in the field of nonlinear optics and may have potential applications in integrated photonics, where efficient nonlinear interaction is expected in a very short interaction distance. Meanwhile, the phase-gradient metasurface is also a very interesting research direction, which has been used by the authors' group to demonstrate novel linear optical effects such as generalized refraction and reflection, mode conversion in waveguides and asymmetric transmission et al in previous publications. In this work they extended the concept to the second-order nonlinear optical effects, and achieved exciting experimental results. Therefore, I recommend its publication in Nature Communications. Below are some technical questions that may help to improve the quality of the paper.

1. It will help readers to better understand the structure of the phase-gradient dielectric arrays if the authors can make the width, length, thickness and distance dz in Fig. 1a, and explain how to calculate $d\phi$ using these parameters, like the mentioned $d\phi=0.5$ degree in Line 112.
2. The mode conversion plays a critical role in the proposed mechanism for effective nonlinear interaction. Therefore, the authors should comment on how the modes should be selected and controlled for the sake of stronger second harmonic generation. In the supplementary information, the authors listed how the second harmonic energy distribute to different higher orders of mode, but they did not comment if these SH modes (mn) appear as they expected, or they just appear randomly by themselves? In other word, is the process controllable? If yes, how the second harmonic mode orders, i.e. m and n should be selected for second harmonic generation.
3. The discussions on the interactions between different orders of second-order harmonics should be transferred from supplementary information to the text in my opinion. From lines 69 to 81, the authors described the physics for the efficient second harmonic generation in their designed metasurface-assisted waveguide. It says the fundamental mode at second harmonic was converted to higher modes. Then it is natural to ask how these second harmonic of different modes interact with each other for the overall conversion efficiency.
4. The authors should introduce briefly the fundamental light source used for the second harmonic generation (pulse or cw, power, polarization, et al) and how the normalized conversion efficiency is defined (normalized to average power or peak power if using pulsed laser). I checked both method and supplementary, but did not find any related information. Please ignore this comment if I ignored something.

Reviewer #2 (Remarks to the Author):

The authors proposed a new scheme to achieve phase-matching-free enhanced nonlinear effect. The utilized structure is a nonlinear waveguide integrated with the dielectric gradient metasurface. The metasurface can introduce an additional wave-vector, leading to a one-way conversion from the zero-order waveguide mode to high-order modes for the second harmonic (SH) signal. Moreover, due to the small overlapping between high-order SH mode and the zero-order first harmonic mode, such nonlinear enhancement effect can be maintained and increase with the total size of the metasurface.

The idea is quite interesting and the paper is well organized. Here, I have some questions/comments for the authors:

1) Although the nonlinear effect is enhanced by the gradient metasurface, the price is the change of the mode order. Besides, multiple high-order modes with distinct polarizations may co-exist in the output ports. These features may be inconvenient in applications. Are there some ideas to solve them? I suggest the authors add some discussion in the revised version.

2) Figure 4 (a) shows a large SH peak at 1667 nm due to the accidental phase matching. Such enhancement is much stronger than that in other wavelength regime. Although the metasurface can also help for further enhancement but not so much. My question is: if the metasurface is re-designed at 1667nm, can the SH signal become stronger assisted by the two effects (accidental phase matching and the modal index manipulation)?

3) The proposed device has a wide band for the enhanced nonlinear effect. On the other hand, gradient metasurfaces also have their individual working bands. In the present manuscript, the authors do not clarify the relationship between the two bandwidths. Is the bandwidth of the enhanced nonlinear effect totally determined by that of the metasurface? What about their detailed relationships? The authors should clarify these points.

4) In Fig.4(a), the SHG signal show some fluctuation. According to the reasons provided by the authors, similar phenomenon should be also observed in simulations. However, we do not see the fluctuation in Fig. 2(c). Why?

5) Here, the metasurface plays an very important role to increase the mode index. In the manuscript, each antenna array consists of 35 a-Si nanostructures with a phase difference of 0.5 degree. I guess that such phase gradient is utilized to match the difference of the mode index between the 0-order and desired high-order modes. And the authors should make clear the design strategy in the manuscript. Besides, if we choose a different phase gradient (larger or smaller), does the similar enhanced nonlinear effect also exist?

I will support this manuscript for the publication in Nature Communications, after the authors have addressed these points.

Reviewer #3 (Remarks to the Author):

This work presents a novel way to build up second harmonic (SH) emission in a waveguide. Dispersion in a waveguide typically does not allow for efficient phase matching, which makes SH generation very inefficient, i.e. SH-emission is generated and reabsorbed in short length intervals rather than build up over the length of the waveguide. The authors propose to combine the waveguide with a gradient metasurface, which would transfer the SH-emission in the TE₀₀-mode to higher order modes, avoiding reabsorption and thus allowing for a build up of the SH-emission.

The authors experimentally demonstrate this system with a LiNbO₃ waveguide combined with a aSi-metasurface. They successfully show how SH-emission is builds up over the length of the metasurface.

This paper is well written and presented, and shows experimentally how gradient metasurface can be used to create an effective SH-waveguide device that doesn't rely on phase-matching. It is in no doubt novel and of high interest to a broad readership. Thus I recommend it for publication in Nature Communication, provided that the following issues are properly addressed:

1. The working principle explained in the first section of Results does not clearly show what limits the efficiency of the device. This is also not clear from the simulations. The authors should consider adding a simple coupled mode theory model to better elucidate contribution of the different coupling processes.

2. It is not stated in the main manuscript what kind of EM-simulations are used. This should be added.

Reply to reviewers' comments

Review #1

In this manuscript, the authors presented both theoretical and experimental research on second harmonic generation in a lithium niobate waveguide assisted with phase-gradient metasurface. A new mechanism of efficient second-order nonlinear optical interaction is demonstrated relying on mode conversions in the phase-gradient dielectric antenna array. This is an exciting progress in the field of nonlinear optics and may have potential applications in integrated photonics, where efficient nonlinear interaction is expected in a very short interaction distance. Meanwhile, the phase-gradient metasurface is also a very interesting research direction, which has been used by the authors' group to demonstrate novel linear optical effects such as generalized refraction and reflection, mode conversion in waveguides and asymmetric transmission et al in previous publications. In this work they extended the concept to the second-order nonlinear optical effects, and achieved exciting experimental results. Therefore, I recommend its publication in Nature Communications. Below are some technical questions that may help to improve the quality of the paper.

Comment #1

It will help readers to better understand the structure of the phase-gradient dielectric arrays if the authors can make the width, length, thickness and distance dz in Fig. 1a, and explain how to calculate $d\phi$ using these parameters, like the mentioned $d\phi=0.5$ degree in Line 112.

Response: *We have updated Fig. 1 to include a magnified view of the antenna array, and to indicate the corresponding structural parameters on the schematic views. We have also added one paragraph in the Supplementary Note 2 and included a new supplementary figure (Supplementary Figure 2) to explain how the antenna phase response is calculated. We have included a summary of the simulation methods in the "Methods" section.*

Comment #2

The mode conversion plays a critical role in the proposed mechanism for effective nonlinear interaction. Therefore, the authors should comment on how the modes should be selected and controlled for the sake of stronger second harmonic generation. In the supplementary information, the authors listed how the second harmonic energy distribute to different higher orders of mode, but they did not comment if these SH modes (mn) appear as they expected, or they just appear randomly by themselves? In other word, is the process controllable? If yes, how the second harmonic mode orders, i.e. m and n should be selected for second harmonic generation.

Response: *In our design, we do not control the generated second-harmonic signal to be selectively coupled into a specific higher-order waveguide mode. As long as it couples into higher-order mode(s), the light at second-harmonic frequency will not be able to couple back into the fundamental frequency, and thus will keep accumulating. In other words, the order and the purity of the converted waveguide mode is not crucial for the phase-matching-free process to happen, which is also one reason that our devices are robust against fabrication imperfections. If needed for certain applications, however, the mode conversion can be well controlled and the purity of converted mode can be high by using a single phased antenna array, as we have shown experimentally in our previous work (Ref. 34). We have added two sentences (Page 7, line 7-9) to elaborate this point.*

Comment #3

The discussions on the interactions between different orders of second-order harmonics should be transferred from supplementary information to the text in my opinion. From lines 69 to 81, the authors described the physics for the efficient second harmonic generation in their designed metasurface-assisted waveguide. It says the fundamental mode at second harmonic was converted to higher modes. Then it is natural to ask how these second harmonic of different modes interact with each other for the overall conversion efficiency.

Response: *The generated second harmonic signal is in a mixture of higher-order waveguide modes with different phases and spatial distributions. They will beat with each other in the course of propagation. As a result, the SHG power fluctuates after it passes through the region patterned with phased antenna arrays, as is shown in Supplementary Figure 4. Indeed, this is partly the reason that the measured SHG spectra are not as flat as in simulations, and is discussed in detail in the last paragraph of “Experimental characterization”. We have added some discussion in the design section (Page 8, lines 17-19) to better illustrate this point.*

Comment #4

The authors should introduce briefly the fundamental light source used for the second harmonic generation (pulse or cw, power, polarization, et al) and how the normalized conversion efficiency is defined (normalized to average power or peak power if using pulsed laser). I checked both method and supplementary, but did not find any related information. Please ignore this comment if I ignored something.

Response: *The detailed characterization setup is described in the second paragraph of the Methods section (now highlighted). In particular, we used a CW telecom tunable laser (max power ~ 20 mW) and used a fiber polarization controller for polarization manipulation. We have added two sentences in the Methods section to explain the definition of normalized conversion efficiency.*

Review #2

The authors proposed a new scheme to achieve phase-matching-free enhanced nonlinear effect. The utilized structure is a nonlinear waveguide integrated with the dielectric gradient metasurface. The metasurface can introduce an additional wave-vector, leading to a one-way conversion from the zero-order waveguide mode to high-order modes for the second harmonic (SH) signal. Moreover, due to the small overlapping between high-order SH mode and the zero-order first harmonic mode, such nonlinear enhancement effect can be maintained and increase with the total size of the metasurface.

The idea is quite interesting and the paper is well organized. Here, I have some questions/comments for the authors. I will support this manuscript for the publication in Nature Communications, after the authors have addressed these points.

Comment #1

Although the nonlinear effect is enhanced by the gradient metasurface, the price is the change of the mode order. Besides, multiple high-order modes with distinct polarizations may co-exist in the output ports. These features may be inconvenient in applications. Are there some ideas to solve them? I suggest the authors add some discussion in the revised version.

Response: *We thank the reviewer for this nice question. The generation of multiple high-order waveguide modes is currently a limitation of our system, and is a compromise in order to achieve the phase-matching-free process. Nonetheless, our devices can still find applications where the output mode purity is not essential. In these cases, the SHG can be collected and guided using a multimode fiber, just as we have done in the current work.*

In principle, the gradient metasurface can be designed to convert the SH light into one specific high-order waveguide mode, or even fundamental (TM) mode, as we have previously shown experimentally in Ref. 34. In this way, after one set of antenna array, the SH light will be converted into a pure high-order mode, while the pump light will remain in the fundamental mode. We can then couple the remaining pump power to an adjacent waveguide and use another phased antenna array to repeat the SHG and mode conversion process. This process can be repeated until the pump is depleted, resulting in many waveguides with SH power all in one specific waveguide mode. The generated SH light can be combined into a single waveguide and/or converted back into fundamental mode using a reversed phased antenna array. This approach will be studied in our future work. We have included a few more sentences in the Discussion section to elaborate on this point (Page 12, lines 7-10).

Comment #2

Figure 4 (a) shows a large SH peak at 1667 nm due to the accidental phase matching. Such enhancement is much stronger than that in other wavelength regime. Although the metasurface can also help for further enhancement but not so much. My question is: if the metasurface is re-designed at 1667nm, can the SH signal become stronger assisted by the two effects (accidental phase matching and the model index manipulation)?

***Response:** We would like to point out that our metasurface is designed to cover a wide pump wavelength range including 1667 nm. The reason that the accidental phase-matched SHG peak at 1667 nm overwhelms the effect from metasurface is because the two processes take place over length scales that are two orders of magnitude different (19 μm for metasurface region and 1.5 mm for the entire waveguide). Within the same nonlinear interaction length, the SHG process assisted by the gradient metasurfaces is at least 3 orders of magnitude more efficient than the accidental phase matching process in a bare LN waveguide. One potential way to further enhance the overall conversion efficiency and to make the output optical mode more controllable is explained in our reply to the previous comment, and is discussed now in the “Discussion” section.*

Comment #3

The proposed device has a wide band for the enhanced nonlinear effect. On the other hand, gradient metasurfaces also have their individual working bands. In the present manuscript, the authors do not clarify the relationship between the two bandwidths. Is the bandwidth of the enhanced nonlinear effect totally determined by that of the metasurface? What about their detailed relationships? The authors should clarify these points.

***Response:** The bandwidth of the enhanced nonlinear effect is indeed the same as the bandwidth of the gradient metasurface. However, the system bandwidth does decrease as a function of the number of antenna arrays (Fig. 2c), which follows the same trend of the bandwidth of gradient metasurface mode convertor. The tolerance of the wavevector Δk provided by the phased antenna array is Fourier-transform limited by the total array length L . As L increases, the range of momentum Δk it can provide decreases. As a result, mode conversion becomes less tolerant to phased antenna design and the bandwidth of the mode conversion will decrease. We have added one sentence to clarify this in the last paragraph of the “Device design and numerical simulations” section (Page 9, lines 2-4).*

Comment #4

In Fig.4(a), the SHG signal show some fluctuation. According to the reasons provided by the authors, similar phenomenon should be also observed in simulations. However, we do not see the fluctuation in Fig. 2(c). Why?

***Response:** As we have stated in the last paragraph of “Experimental characterization” section, the SHG signal fluctuation comes from mode interactions after light propagates through the metasurface*

region. The simulation results in Fig. 2 only show the device response monitored right after the metasurface region. The explanation for the oscillatory behavior of SHG signal is supported by numerical simulations provided in Supplementary Figure 4. Another reason is that, in simulations we collected the power from all generated higher-order waveguide modes, while in experiments the fiber collecting efficiency depends on the mode profile. As a result, different combinations of higher-order modes as a function of the pump wavelength will result in different collected power levels in experiments. This is also discussed in the last paragraph of “Experimental characterization” section. We have highlighted the corresponding discussions.

Comment #5

Here, the metasurface plays a very important role to increase the mode index. In the manuscript, each antenna array consists of 35 a-Si nanostructures with a phase difference of 0.5 degree. I guess that such phase gradient is utilized to match the difference of the mode index between the 0-order and desired high-order modes. And the authors should make clear the design strategy in the manuscript. Besides, if we choose a different phase gradient (larger or smaller), does the similar enhanced nonlinear effect also exist?

Response: *As we have discussed in the reply to reviewer #1, we do not control the generated second harmonic light to be selectively coupled into a specific higher-order mode. As long as it couples into higher-order mode(s), the light at second harmonic frequency will not be able to couple back into the fundamental frequency, and thus will keep accumulating. In other words, the order and the purity of the converted mode are not crucial for the phase-matching-free process to take effect, which is also one reason that our devices are robust against fabrication imperfections. If needed for certain applications, however, the mode conversion can be well controlled and the purity of the converted mode can be high by using a single phased antenna array, as we have shown experimentally in our previous work (Ref. 34). We have added two sentences (Page 7, lines 7-9) to elaborate this point.*

We have also added one paragraph in the Supplementary Note 2 and included a new supplementary figure (Supplementary Figure 2) to explain how the antenna phase response is calculated. We have included a summary of the simulation methods in the “Methods” section.

The phase gradient should be within some range, not too small or too large. If the phase gradient is too small, the mode conversion is slow, and thus there could be unwanted power beating between the SH signal and the pump. On the other hand, if the phase gradient is too large, the SH signal will quickly couple into waveguide modes of very high order, and will eventually leak out from the waveguide after few antenna arrays.

Review #3

This work presents a novel way to build up second harmonic (SH) emission in a waveguide. Dispersion in a waveguide typically does not allow for efficient phase matching, which makes SH generation very inefficient, i.e. SH-emission is generated and reabsorbed in short length intervals rather than build up over the length of the waveguide. The authors propose to combine the waveguide with a gradient metasurface, which would transfer the SH-emission in the TE₀₀-mode to higher order modes, avoiding reabsorption and thus allowing for a build up of the SH-emission.

The authors experimentally demonstrate this system with a LiNbO₃ waveguide combined with a aSi-metasurface. They successfully show how SH-emission is builds up over the length of the metasurface.

This paper is well written and presented, and shows experimentally how gradient metasurface can be used to create an effective SH-waveguide device that doesn't rely on phase-matching. It is in no doubt

novel and of high interest to a broad readership. Thus I recommend it for publication in Nature Communication, provided that the following issues are properly addressed:

Comment #1

The working principle explained in the first section of Results does not clearly show what limits the efficiency of the device. This is also not clear from the simulations. The authors should consider adding a simple coupled mode theory model to better elucidate contribution of the different coupling processes.

Response: *Our device represents a complicated system containing many (>5) waveguide modes coupled through the gradient metasurface. While the phased-antenna array converts TE_{00} mode at the SH frequency into higher-order TE/TM modes, it also further converts these modes into even higher-order TE/TM modes. This continuous flow of optical power from low- to high-order modes sets the ultimate conversion efficiency limitation of the device, since the light will eventually be converted into leaky modes and lost into the substrate. We have added one sentence at the end of the “Theoretical principle of the phase-matching-free nonlinear process” section to better explain this (Page 6, lines 2-3).*

Comment #2

It is not stated in the main manuscript what kind of EM-simulations are used. This should be added.

Response: *The numerical simulation is conducted using Lumerical FDTD solutions. The detailed simulation methods are provided in the Supplementary Note 2. We have added one sentence in the main text to cite this information. We have also included a summary of the simulation methods in the “Methods” section.*

List of Modifications

* All new text has been highlighted in YELLOW in the revised manuscript.

1. In Page 6, lines 2-3: A new sentence has been added to discuss the ultimate limitation on the metasurface-assisted nonlinear process. The new text reads: *“This sets the ultimate limitation on the highest nonlinear conversion efficiency achievable in the current device scheme.”*
2. In Page 7, lines 8-10: New text has been added to discuss the choice for phase gradient in the antenna arrays. The new text reads: *“Note that the actual value of $d\phi$ is not critical for the phase-matching-free process to take effect. As long as the antenna array converts SH light into higher-order waveguide mode(s) at a rate comparable to the SHG process, the total SH signal will keep accumulating.”*
3. In Page 8, lines 17-19: A new sentence has been added to discuss the SHG intensity variations after the metasurface-patterned section. The new text reads: *“The interference between these higher-order modes after the metasurface-patterned section will cause intensity variations of the generated SHG signal, but the overall SHG power would stay on the same level as that right after the antenna arrays (Supplementary Figure 4).”*
4. In Page 9, lines 2-4: A new sentence has been added to explain the SHG bandwidth change as the number of antenna arrays increases. The new text reads: *“The SHG enhancement bandwidth decreases, however, as the number of antenna arrays increases, since a longer interaction distance corresponds to less tolerance on the phase gradient of the antenna arrays.”*

5. In Page 11, lines 8-16: The explanation on why the measured SHG spectra are not as flat as in simulations is highlighted for reviewer #2 (Comment #4).
6. In Page 12, lines 7-10: New text has been added to discuss the drawback of the current approach and possible way to solve the problem. The new text reads: *“One drawback of our current approach is the difficulty of efficient light collection from the generated higher-order waveguide modes. This could possibly be solved by using a single antenna array to convert the SH light into one specific mode (e.g. TM₀₀)³⁴ and extracting the remaining pump light into an adjacent waveguide to repeat the process.”*
7. In Page 12, a new paragraph has been added (first paragraph) to the Methods section to include the numerical simulation methods used (previously in the Supplementary Information). A new sentence has also been added in Page 7, line 12 to refer to this section.
8. In Page 13, lines 12-14: the description on the light sources used for SHG measurement is highlighted for reviewer #1 (comment #4).
9. In Page 13, lines 17-21: new text has been added to describe the definition for normalized SHG conversion efficiency. The new text reads: *“The input and output power used for conversion efficiency calculation refer to the on-chip power into and out of the metasurface region, taking into consideration the fiber-to-chip coupling losses. The normalized conversion efficiency is defined as $\eta = P_{out}/(P_{in}^2 \cdot L^2)$, where P_{in} and P_{out} are the input and output power, respectively, and L is the length of the metasurface region.”*
10. In Supplementary Information, new text and a new figure (now Supplementary Figure 2) has been added to explain in detail the design methods for the phased-antenna array, and to include the information on refractive indices and crystal directions used in simulations.
11. The order of Supplementary Notes and Figures has been adjusted to follow the callout order in the main text.
12. The main text and supplementary information have been proofread to correct typos and grammar errors.

REVIEWERS' COMMENTS:

Reviewer #1 (Remarks to the Author):

I agree the revisions that had made by the authors, and therefore I recommend its publication in Nature Communications.

Reviewer #2 (Remarks to the Author):

The authors have well addressed all of my comments. I have no more questions and support its publication in present form.

Reviewer #3 (Remarks to the Author):

Considering the responses to the reviewers comments and the changes made in the text, I can recommend the manuscript for publication in Nature communication.

Response to reviewers' final remarks

Reviewer #1 (Remarks to the Author):

I agree the revisions that had made by the authors, and therefore I recommend its publication in Nature Communications.

Response: We thank the reviewer for his/her positive feedback.

Reviewer #2 (Remarks to the Author):

The authors have well addressed all of my comments. I have no more questions and support its publication in present form.

Response: We thank the reviewer for his/her positive feedback.

Reviewer #3 (Remarks to the Author):

Considering the responses to the reviewers comments and the changes made in the text, I can recommend the manuscript for publication in Nature communication.

Response: We thank the reviewer for his/her positive feedback.